# Echocardiographic Evaluation in Paediatric Sickle Cell Disease Patients: A Pilot Study

**DOI:** 10.3390/jcm12010007

**Published:** 2022-12-20

**Authors:** Letizia Sabatini, Marcello Chinali, Alessio Franceschini, Margherita Di Mauro, Silvio Marchesani, Francesca Fini, Giorgia Arcuri, Mariachiara Lodi, Giuseppe Palumbo, Giulia Ceglie

**Affiliations:** 1Department of Pediatrics, Bambino Gesù Children’s Hospital—IRCCS, Piazza Sant’Onofrio 4, 00165 Rome, Italy; 2Department of Systems Medicine, University of Rome Tor Vergata, Viale Oxford 81, 00133 Rome, Italy; 3Department of Pediatric Cardiology and Cardiovascular Surgery, Bambino Gesù Children’s Research Hospital—IRCSS, Piazza Sant’Onofrio 4, 00165 Rome, Italy; 4Department of Pediatric Hematology and Oncology, Cell and Gene Therapy, Bambino Gesù Children’s Hospital—IRCCS, Piazza Sant’Onofrio 4, 00165 Rome, Italy

**Keywords:** sickle cell disease, paediatric population, cardiovascular complications, echocardiography

## Abstract

Cardiovascular involvement has a great impact on morbidity and mortality in sickle cell disease (SCD). Currently, few studies are available regarding the paediatric setting and, moreover, current guidelines for the echocardiogram screening program in the asymptomatic paediatric population are controversial. We performed a retrospective observational monocentric study on 64 SCD patients (37 male and 27 female, median age 10) at the Bambino Gesù Childrens’ Hospital, who had undergone a routine transthoracic echocardiogram. In total, 46 (72%) patients had at least one cardiac abnormality. Left atrial dilatation (LAD) was present in 41 (65%) patients and left ventricular hypertrophy (LVH) was found in 29 (45%) patients. Patients with LAD showed lower median haemoglobin levels (*p* = 0.009), and a higher absolute reticulocyte count (*p* = 0.04). LVH was negatively correlated with the median haemoglobin value (*p* = 0.006) and positively with the reticulocyte count (*p* = 0.03). Moreover, we found that patients with cardiac anomalies had higher transfusion needs and a lower frequency of pain crises. In our setting, cardiac involvement has a high prevalence in the paediatric cohort and seems to be associated with specific laboratory findings, and with a specific clinical phenotype characterized by complications related to high haemodynamic load.

## 1. Introduction

Sickle cell disease (SCD) is an inherited blood disorder caused by a mutation in the gene encoding the haemoglobin subunit β. This term includes a group of autosomal recessive diseases affecting red blood cells: sickle cell anaemia is the most common, occurring in 60–75% of patients, characterized by homozygous mutant haemoglobin S (HbSS); about 25% of patient have compound heterozygosity of haemoglobin S with other β-globine chain variants or β-thalassemia traits [1]. The sickle Hb allele (HbS) is characterized by a single adenine-to-thymine mutation which results in glutamate being substituted by valine at position 6 in the mature beta-globulin chain [2]. This substitution modifies the quaternary form of the protein at a low oxygen concentration, leading to HbS polymerisation in elongated rope-like fibres [3]. The precipitation of HbS tetramers within red blood cells results in a marked decrease in cell deformability and in a change in cell morphology, resulting in the sickle shape from which the disease takes its name. Although initially reversible, repeated episodes of sickling during deoxygenation can irreversibly damage the cell membrane, until cells fail to return to their normal biconcave disc shape when normal oxygen tension is restored [4].

The estimated prevalence is about 1 in every 360 new-borns among African Americans and the number of SCD cases is expected to increase in the next decades [5]. The disease is more common in geographical areas where malaria is endemic, such as sub-Saharan Africa, India, Middle East and some Mediterranean regions. The most supported reason for this is the so-called “malaria hypothesis”, according to which the presence of the HbS mutation might provide a protection against Plasmodium Falciparum infections, making HbS carriers more resistant, therefore conferring a selective advantage [6]. Nevertheless, because of migratory flow, SCD is becoming more and more common in Western regions and countries, such as Europe and the United States, making it a social and health problem of global significance, with geographical heterogeneities in prevalence and life expectancy [7].

The introduction of new therapies, an active surveillance program, a more extensive vaccination coverage during childhood, and new-born screening have determined a great improvement in the clinical management of SCD patients, resulting in an increase in the average life expectancy of patients [8]. Thus, the medical burden of chronic complications has become heavier than in the past, requiring novel diagnostic and therapeutic strategies.

The clinical picture of SCD is drawn by two main phenomena: vaso-occlusive pain crisis and chronic haemolytic anaemia, which can be differently expressed and prevalent in different subjects [5]. HbS polymerisation is the key event leading to the formation of stiff, sticky, sickle-shaped erythrocytes that can obstruct microcirculation, resulting in blood flow occlusion, tissue ischemia and pain crisis [9]. Sickled red blood cells are also subject to haemolysis [10]. Hypoxia, acidosis and inflammation are known triggers of this plethora of events that involve not only red blood cells but also leucocytes, platelets and endothelial cells. This cascade of events promotes a vicious cycle that builds on and reinforces itself until appropriate therapeutic measures are applied [11].

The recurrence over time of acute occlusive crisis and haemolytic events induces cumulative end-organ complications and failure, affecting mostly the cardiovascular system, kidneys, central nervous system and bone tissue. Among those, cardiopulmonary dysfunction has a great impact on morbidity and premature mortality. The most common abnormalities found in SCD patients are left atrial dilatation and increased left atrial mass, right ventricular and left ventricular enlargement, diastolic dysfunction and pulmonary hypertension [12], which involve up to 82% of adult patients [13]. Chronic anaemia has been considered the main factor underlying these complications: increased cardiac output to ensure an adequate tissue oxygen supply is associated with cardiac chamber dilatation, chronic ventricular wall stress and compensatory hypertrophy. New research over the last decade has revealed that intravascular haemolysis also gives an independent contribution to the pathogenesis of vascular injury and organ disfunction [14]. In fact, free plasma Hb and haem inhibit NO signalling, promoting reactive oxygen species (ROS) formation, leading to an impairment in the vascular redox state and in the balance between vasodilator and vasoconstrictor factors [15]. This results in an increase in vascular resistance, local inflammation, endothelial dysfunction, hypercoagulation state and proliferative vasculopathy, with consequent local ischemia and tissue damage. Within the pulmonary microcirculation, these events have an important role in the development of pulmonary hypertension (PH), one of the leading cause of premature death in SCD patients [16].

Echocardiography is a useful non-invasive screening method for cardiac involvement and PH. The tricuspid regurgitation jet velocity (TRV) value, measured via Doppler echocardiography, provides an indirect assessment of right ventricular and pulmonary artery systolic pressures, and a value TRV >/= 2.5 m/s is associated with a high risk for mortality and a clinal diagnosis of PH, that needs to be confirmed by right ventricle catheterization [17]. According to the recent consensus guidelines from the American Society of Haematology published in 2019, routine echocardiogram (ECHO) screening to identify PH is not recommended in asymptomatic children and adults, without any comorbid conditions or disease complications associated with PH [18].

In fact, considering the high impact of cardiac complications in life quality and expectancy, it would be of the utmost importance to detect cardiac abnormality early in childhood. So far, few studies have evaluated cardiovascular status in the paediatric SCD population [13,19]. Thus, our study aims to investigate the impact of cardiac involvement in a paediatric SCD cohort and the role of clinical and laboratory markers as predictive factors to identify patients at a higher risk of cardiac complications.

## 2. Materials and Methods

### 2.1. Population Study

This is a retrospective observational monocentric study on a paediatric SCD population currently followed at the Bambino Gesù Childrens’ Hospital. All the patients included underwent a routine transthoracic echocardiogram during follow up. The echocardiogram was performed during a patient’s steady-state, regardless of the presence of cardiological symptoms, according to a screening protocol for the evaluation of cardiovascular involvement applied in our centre. We collected personal information, medical history, and laboratory data referring to the 2 years prior to the echocardiogram from the hospital’s database of electronic medical records.

### 2.2. Echocardiographic Evaluations

All patients underwent complete transthoracic echocardiographic examination with commercially available machines (Epiq7 or iE33, Philips Heathcare Inc., Andover, MA, USA). Examinations were analysed off-line on dedicated workstations (QLab Cardiac Analysis ver.10, Philips Heathcare Inc.) by two expert independent readers blinded of the clinical data (AF, MC). Two-dimensional images were obtained for the analysis of LV volumes on three consecutive beats from apical 4- and 2-chamber views. Wall thickness and chamber dimensions were obtained from the two-dimensional parasternal long axis or M-mode short axis at the midventricular level, when perfect alignment of the left ventricle was possible, on three consecutive beats. Parameters measured in our study included LV diameters and wall thicknesses to obtain left ventricular mass (LVM). LV mass was calculated according to Devereux’s formula [20]. LV mass was analysed both as unindexed and as indexed by age specific allometric powers. To define LV hypertrophy, a partition value of 45 g/(m^2.16+0.09^) was used [21], as recently suggested by the European consensus on paediatric hypertension [22]. For the evaluation of LV geometry, myocardial thickness (wall + septum) was divided by the LV minor axis (diameter) to generate a relative wall thickness (RWT) normalized to a mean age of 10 years old. A value of 0.38 was used as the cut-off to define concentric LV geometry [23]. Left ventricle dilation was defined by LV diameter mm/BSA derived Z scores [24]. LV systolic function was determined by the LV ejection fraction (EF) calculated using the biplane Simpson formula [25]. Normal EF was defined according to current adult guidelines with a cut-off of 55%. Left atrial dilation was defined by LA volume/BSA^1.48^ derived Z scores [26]. Standard Doppler analysis was performed to obtain LV inflow velocities at the mitral valve tips, including peak early diastolic filling (E) and late diastolic peak velocities (A). Tissue Doppler analysis was performed to obtain longitudinal early diastolic (E′) velocities at both the septal and the lateral in order to derive the mitral E/e′ ratio [25]. Strain analysis was used to obtain the regional and global longitudinal strain (GLS) of the left ventricle on three consecutive beats from each apical window (apical 4-chamber, 3-chamber and 2-chamber window), comprising a total of 9 beats for each GLS analysis [27]. Averaged segmental peak strains were calculated to obtain the regional and global longitudinal strain. The prevalence of subclinical cardiac systolic dysfunction was defined by previously reported age-specific strain partition values obtained from a meta-analysis of over 1100 children, with impaired GLS defined as GLS > −20.5% in age range 2–9; GLS > −19.1% in age range 9–13; and GLS > −19.2% in age range 14–21 [28].

### 2.3. Clinical Evaluation

All patients underwent periodic medical examinations and serial blood chemistry tests during follow-up including: complete blood count, haemoglobin assay using the HPCL method, liver function test (AST, ALT, LDH, GGT, total and conjugated bilirubin), screening markers for iron accumulation (ferritin), inflammatory markers (CRP), markers of renal and hepatic damage (Urea Nitrogen, Creatinine and Uric Acid) and markers of cardiac overload (aminoterminal fragment of the natriuretic propeptide type B, NT-proBNP). The relevant clinical data considered were the number of painful crises and blood transfusions received over the previous two years and ongoing hydroxyurea therapy. Regarding the main complications of the disease, the presence of acute thoracic syndrome, osteomyelitis, leg ulcers, cholelithiasis or cholecystectomy were considered during the same time period. In addition, patients were also analysed based on splenic function considering that SCD patients may undergo a splenectomy in the course of the pathology, or progress to “functional asplenia”. Moreover, transcranial Doppler ultrasound (TCD) has been performed as screening tool for cerebrovascular disease in children with HbSS or HbSb0 thalassemia (ages 2–16 years) according to the recent ASH guideline [29].

### 2.4. Statistical Analysis

All patients’ data were reported in a password-protected Excel spreadsheet (Microsoft). Continuous variables are presented as mean ± SD or as medians with interquartile ranges (IQR, 25th–75th percentile), as appropriate. Categorical variables are described using counts and percentages. The comparison study between continuous variables was made by the T-Student test (two-sided) for parametric distribution or the Mann–Whitney test for nonparametric distribution. To assess whether the distribution of the variable corresponded to a normal distribution, the Kolmogorov–Smirnov test of normality was carried out using Social Science Statistics software. The comparison between qualitative variables was carried out with the chi-square (χ²) statistical test, applying the Yates correction when necessary. Statistical significance was set at values of *p* ≤ 0.05.

## 3. Results

### 3.1. Population Study

Out of the 82 SCD patients followed in our hospital, a total of 64 patients affected by SCD (37 males and 27 females, mean age 10 years, medium age 9 years with aninterquartile range 5–14 years) have undergone transthoracic echocardiogram in our centre during the last two years (Table 1). The remaining patients underwent the exam externally for various personal reasons and thus were not included in this study because of a lack of data homogeneity. In one patient, it was not technically possible to assess the presence of atrial dilation, so for this parameter the population studied was considered to be 63 patients.

In our cohort, the most common SCD genotype was HbSS (81%), followed by HbSβ thalassemia (9%), HbSC (5%) and other (5%). The demographic origin of the population was heterogeneous, with a predominance of subjects from Nigeria. For each patient, an average of values of laboratory data was reported over the past two years, prior to echocardiographic examination. In order to reflect the patients’ overall clinical situation more realistically, values during acute events were also included.

A total of 35 patients (aged 2–16 years) of the study population (55%) have undergone TCD screening in our centre, which showed a conditional time-averaged mean of the maximum velocity (TAMMV), 170–199 cm/s, in 7 patients and an abnormal TCD measurement, ≥200 cm/s, in 2 cases. The remaining eligible patients performed the exam externally for various personal reasons and, thus, this data has not been considered because of a lack of technique and measurement homogeneity [30].

In this cohort, 45 patients (70%) were taking hydroxyurea (HU): according to the latest guidelines we started the therapy at diagnosis. In almost all cases, hydroxyurea has been started from one year of age, with an initial dose of HU of 10 mg/kg/day for 4 weeks, gradually titrating up to a mean value of 30–35 mg/kg/day, according to bone marrow toxicity and HbF response. A significant number of our patients has been referred to our centre after being diagnosed in another hospital, so in that case the data about the exact duration of HU therapy is missing. Regarding blood transfusions, 28 patients (44%) were receiving chronic transfusion therapy: 2 of them for primary stroke prevention, 2 for secondary stroke prevention, and the remaining 24 patients for severe complications of the disease, especially high-grade haemolytic anaemia. In the median amount of total blood transfusions performed in this cohort, transfusions during acute complications of the disease were also considered.

In addition, four patients from our cohort underwent haematopoietic stem cell transplantation, but echocardiographic examinations performed before the start of conditioning therapy were selected for the study.

### 3.2. Echocardiographic Study

In our cohort of 64 patients, at least one cardiac abnormality was found in 72% of the population. The echocardiographic parameters and measurements of the cohort are described in Table 2.

Left atrial dilation and left ventricular hypertrophy were the most observed changes. Left atrial dilatation (LAD) was present in 41 (65%) patients and left ventricular hypertrophy (LVH) was found in 29 (45%) patients. A total of 25 patients with LAD (61%) also had left ventricular hypertrophy. Only 5 patients had systolic disfunction defined by strain echocardiography, and no patient had indirect signs of pulmonary hypertension, nor diastolic dysfunction.

There was a higher prevalence of males in the population with atrial dilatation (59% male) and in the population with left ventricular hypertrophy (62% male). Concerning ventricular parameters, to normalize sex differences the mass index was used, which corrects the left ventricular mass for body size. No correlations between different age groups and cardiac abnormalities were found.

Patients with LAD showed a lower median Hb level (*p* = 0.009) and a higher absolute reticulocyte count (*p* = 0.04). See Figure 1.

LVH was negatively correlated with the median Hb value (*p* = 0.006) and positively with the reticulocyte count (*p* = 0.03), as seen in Figure 2. Regarding the other main markers of haemolysis (lactate dehydrogenase, unconjugated bilirubin, aspartate transaminase), both patients with LAD and LVH presented a slight tendency to have increased values, without a significant correlation.

See Table 3 and Table 4 for clinical and laboratory data of study groups based on echocardiography abnormalities.

The presence of these cardiac alterations was not found to be correlated with genotype nor with hydroxyurea therapy.

Regarding splenic function, in our population, 7 patients had undergone surgical splenectomy, 11 patients had a condition of “functional asplenia” or “auto-splenectomy”, defined as a complete or partial decrease in tissue uptake on Tc-99m red blood cell scintigraphy. The presence of sustained hypersplenism and life-threatening episodes of acute splenic sequestration have been the primary indication of splenectomy in our cohort. Considering these two subgroups together, we found a statistically significant correlation between left ventricular hypertrophy and impaired spleen function (*p* = 0.03) due to both surgical splenectomy and auto-splenectomy.

Among the other clinical-anamnestic parameters considered, we did not find any significant correlation with cardiac alteration and the main complication of the disease analysed.

Interestingly, we found that patients with cardiac anomalies had higher transfusion needs and a lower frequency of pain crises, although statistical significance was not achieved (*p* = 0.090 in the group with atrial dilation and *p* = 0.096 in the group with ventricular hypertrophy).

## 4. Discussion

Cardiac complications are a common feature of SCD and they represent an important cause of morbidity and mortality. In fact, left ventricular diastolic dysfunction and pulmonary hypertension measured by echocardiogram are independent predictors of death in adults [20]. Growing evidence support that children with SCD are also affected by cardiac complications. Despite this, the evidence-based guidelines of the American Society of Haematology [18] do not recommend routine echocardiographic (ECHO) screening to identify PH in asymptomatic children and adults. In this study, conducted on 64 patients with SCD at Bambino Gesù Children Hospital, we analysed the echocardiographic data and their correlation with clinical and laboratory data. Among the main results obtained, we stress the important prevalence of left atrial dilation and left ventricular hypertrophy in paediatric age, but no abnormalities in pulmonary pressure and/or in functional parameters.

Since the onset of cardiovascular complications occurs progressively in the course of the disease, their incidence increases with patient age and is higher in adulthood [31]. In this study, although we could not find a statistically significant difference, subjects without anomalies were younger (mean age of 9 years) than patients with dilation or hypertrophy, in which a mean age of 11 years has been calculated. Considering gender-related differences, our study showed that males tend to be the most frequently affected by cardiac complications, as already documented in the literature [32,33].

Left atrial dilation was found in 65% of our study population. This finding is comparable to what is described in other studies on paediatric cohorts [34,35]. This cardiac alteration is known to be a good marker of diastolic dysfunction and a predictor of cardiac events in the general population [31,36]. Left ventricular hypertrophy was found in a greater number of children compared to other studies in the literature [32,37,38]. Significantly, the study shows a slight prevalence of atrial dilation compared to ventricular hypertrophy. This suggests that the specific chronological sequence of cardiac adaptation through alterations occurs in a well-determined temporal order. Initially, atrial dilatation develops due to volume overload, leading to a secondary left ventricular hypertrophy. In response to ventricular dilation due to chronic volume overload, the left ventricle adapts to a condition of greater wall stress by increasing its mass and thus developing compensatory hypertrophy [39]. The thickening of the walls and an elongation of the myofibers allows for a haemodynamic adaptation to initially preserve the diastolic function and the filling pressures [16]. Nevertheless, over time the hypertrophy is maladaptive and leads to overt cardiac dysfunction; nonetheless, the precise onset of dysfunctional events following this alteration is not yet well defined [32,40] and may take some time to develop.

In the adult population with SCD, diastolic dysfunction is estimated to be present in about 20% of individuals and it is a predictive index of a worse clinical course and early death [31,41]. On the other hand, its role in the paediatric population is not well known [13]. In our population, the signs of diastolic disfunction were not found in any patient, confirming the results obtained in the study conducted by Allen et al. [19]. Systolic dysfunction in the literature is also described with variable prevalence and in our study it was observed in only five patients [42]. Diastolic disfunction is strictly linked with pulmonary hypertension and Doppler echocardiography provides an estimation of high right ventricular systolic pressure by measuring tricuspid regurgitation jet velocity (TRV). An elevated TRV represents a marker of possible PH and must be confirmed by right heart catheterization (RHC). In our cohort, we did not find any patients with high TRV, differently from other studies which observed an elevation of TRV with variable prevalence [19,43]. In the research conducted by Allen et al., 172 subjects between the ages of 10 and 23 years were included, thus a population older than ours. This difference may again suggest the importance of the time factor for the onset of this complication.

An elevated serum NT-pro-BNP measurement (≥160 pg/mL), which assess ventricular strain, represents an independent predictor of mortality in SCD patients [44]. The use of NT-pro-BNP alone as a screening for PH has not been studied, although in association with TRV, it appears to increase the predictive capacity for PH [45]. In our cohort, we find a slightly increased level of NT-pro-BNP in patients with cardiac disease (LA dilation and LV hypertrophy), although not statistically significant.

Linking echocardiographic findings with laboratory parameters, we found that patients with cardiac anomalies, both atrial dilatation and ventricular hypertrophy, had lower haemoglobin values and higher reticulocyte counts with statistical significance. From a pathophysiological point of view, chronic anaemia implies an increase in cardiac output in compensatory function to meet the oxygen needs of peripheral tissues. Chronic volume overload leads to a progressive cardiac remodelling with left atrium dilatation and eccentric hypertrophy of the ventricle in response to increased wall stress [31,42]. It is well known that chronic anaemia plays a major role in the develop of not only cardiac complications but also other chronic SCD-related disease, such as growth failure and leg ulcers [46,47]. In our population, none of the patients presented a history of leg ulcers, whose prevalence is low before 10 years of age, the mean age of our cohort [48].

Over the past decade, growing evidence supports the independent role of intravascular haemolysis, beyond the severity of the anaemia, in the pathogenesis of vascular injury and other main complications of SCD. In fact, the release of intraerythrocytic haemoglobin in the bloodstream promotes endothelial dysfunction through the imbalance of vascular redox pathways, in particular NO signalling, leading to an excessive production of reactive oxygen species (ROS) [14]. Moreover, circulating haemoglobin and haem can trigger innate immune pathways, acting as erythrocyte danger-associated molecular proteins (eDAMPs) which are recognized by Toll-like receptors on the immune cells surface [15,49]. This sterile inflammation further propagates the oxidative stress over endothelial cells and stimulates platelet activation and a hypercoagulation pathway, which contributes to vascular injury and vasculopathy complications [50,51]. Concerning cardiovascular alterations, some studies suggest that there might be a significant correlation between markers of haemolysis and chamber sizes, ventricular mass and pulmonary hypertension [14]. In our population, we observed higher mean and median values of markers of haemolysis (lactate dehydrogenase, unconjugated bilirubin, aspartate transaminase) in both patients with LAD and LVH, although we could not find a statistically significant difference. Among the haemolytic index, reticulocyte count, which is the expression of a compensatory response of the bone marrow to haemolysis, was directly and significantly correlated with the prevalence of cardiac abnormality in our study, in line with what Zilberman et al. have described regarding left ventricular dilatation [13]. In the literature, the increase in absolute reticulocyte count has been linked with other acute and chronic complications of SCD in paediatric age patients, such as cerebral vasculopathy (stroke, increased trans-cranial Doppler velocity, silent stroke) [52,53], hospitalizations for splenic sequestration, vaso-occlusive crisis and early death [54,55].

The phenotypic variability among individuals with SCD is well known and described in the literature. Kato et al. proposed in a recent review the distinction between the four main subphenotypes, based on clinical expression, laboratory data, short and long-term outcomes: vaso-occlusive type, haemolysis and vasculopathy type, high HbF type and pain type [5]. Low haemoglobin, high reticulocyte count, and high haemolytic markers are the main laboratory hallmarks of the haemolysis and vasculopathy SCD subphenotype. Showing a lower haematocrit than patients with vaso-occlusive phenotype, those individuals might have higher transfusion needs and are at risk of pulmonary complications, cerebral vasculopathy, nephropathy and cardiac remodelling due to chronic anaemia. In our population, we found that patients with left atrial dilatation and left ventricular hypertrophy show a tendency to have greater transfusion needs compared with those who have no cardiac abnormality. This finding seems to provide further data in favour of this theory.

Among chronic complications of the disease, splenic hypofunction is nearly constant in sickle cell anaemia, and it is usually associated with splenomegaly. Splenomegaly results from the progressive trapping of sickled red cells in the red pulp, where the slow and open microcirculation favours in vivo deoxygenation and consequent sickling, leading to the adhesion of RBC to the spleen matrix and to spleen macrophages [56]. Repeated sequences of vaso-occlusion, followed by ischaemia, leads to progressive fibrosis and atrophy, resulting in “auto-splenectomy”. According to Rogers et al., splenic dysfunction begins early in the course of the disease, in fact before 12 months of age in 86% of SCA infants assessed by mTc sulphur-colloid liver spleen scintigraphy [57]. Surgical splenectomy is usually reserved for patients who suffer from life threatening episodes of acute splenic sequestration or as a consequence of sustained hypersplenism. Although the main concern about loss of splenic function is infection risk, according to recent evidence, asplenia, notably following surgical splenectomy, has been associated with vascular complications, above all thromboembolic events and pulmonary arterial hypertension (PAH) [58,59,60,61]. The underlying mechanisms are remarkably complex, but it has been assumed that the absent splenic filter allows particulate matter, damaged, adherent or sickled cells and procoagulant cell-derived microparticles to circulate [62], determining an injury and the activation of the endothelium. In our cohort, we found an interesting association between splenectomy and left ventricular hypertrophy, which has never been described in the literature. Thus, further investigations are needed to understand the underlying pathophysiological mechanisms. Moreover, according to recent evidence, we believe that to better characterized spleen function of patients with neither surgical asplenia nor spleen scintigraphic hypocaptation, more specific analysis is needed regarding circulating erythrocyte and leucocyte cells [63].

Cardiac complications are also well described in other haemoglobinopathies. In β-thalassemia major, cardiomyopathy is the primary determinant of prognosis and survival [64] and iron overload and damage is the main cause, secondary to lifelong blood transfusions, extravascular haemolysis, and increased gastrointestinal iron assimilation [65,66]. Growing evidence shows that transfused patients with SCD have a conspicuous minor degree of iron overload-related organ injury compared to thalassemia patients [67]. Notably, in SCD, a major role is played by the damage caused by pro-inflammatory mechanisms that trigger pathways of tissue fibrosis [35]. To explore this area in this study, we analysed the correlation between ferritin level and cardiac abnormality, but we could not find any significant association, in line with the above-mentioned hypothesis. However, ferritin level is only a useful screening tool, but the gold standard to assess myocardial iron overload is the T2* cardiovascular magnetic resonance (CMR), so further studies are needed to confirm this thesis.

This study should be interpreted considering its limitations. First, patients have been evaluated with echocardiography at different stages during the course of the disease; therefore, the population was heterogeneous in terms of the impact of complications, and duration of therapies. Another limitation is the small number of patients enrolled, mostly due to the unicentricity of our study. Moreover, as an observational and retrospective study, it could not show the progression of cardiac alteration with patient age, as it is well known in the literature. The retrospective design of our study does not allow for the estimation of the long-term effects of chronic anaemia and haemolysis on the incidence of growth failure and other complications occurring later in adulthood, such as pulmonary hypertension, leg ulcers and retinopathy. Thus, it would be interesting to continue this research in the future with a prospective study to assess specific longitudinal changes and to study the impact of specific disease-modifying therapies.

## 5. Conclusions

In conclusion, this study analyses the prevalence of echocardiographic cardiac abnormality in paediatric sickle cells patients, showing a significant impact of cardiac alterations already in childhood, although current guidelines do not prescribe routine echocardiographic monitoring in asymptomatic paediatric patients. It was also observed that cardiological abnormalities are closely related to alterations in specific haematological parameters, and to a very specific clinical phenotype characterized by significant chronic haemolysis, severe anaemia, high transfusion needs, and other complications related to high haemodynamic load. This result, considering the limited data in the current literature, encourages further research to detect clinical and laboratory markers that can help to identify, in the paediatric population, the patients at greatest risk. We suggest that performing serial echocardiographic examinations may lead to an early diagnosis of cardiac complications and may help to improve clinical care in order to reduce the cardiovascular morbidity and mortality that burden the population of patients with sickle cell disease.

## Figures and Tables

**Figure 1 jcm-12-00007-f001:**
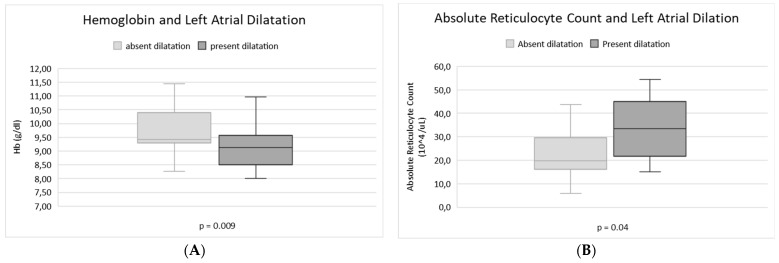
Correlation between LAD and markers of haemolytic anaemia. Box plot showing lower median Hb level (**A**), increased absolute reticulocyte count (**B**) in patients with left atrial dilatation compared to patients without this echocardiography alteration.

**Figure 2 jcm-12-00007-f002:**
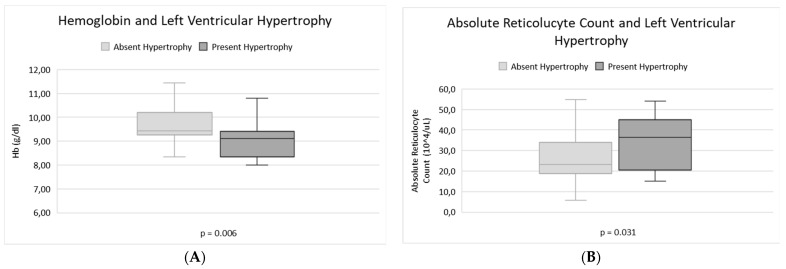
Correlation between LVH and markers of haemolytic anaemia. Box plot showing lower median Hb level (**A**), and increased absolute reticulocyte count (**B**) in patients with left ventricular Hypertrophy compared to patients without this echocardiography alteration.

**Table 1 jcm-12-00007-t001:** Clinical characteristics of the cohort (*n* = 64).

Medium age in years at the time of echocardiography study (IQR)	9 (5–14)
Gender	
Male	37 (58%)
Female	27 (42%)
Genotype	
HbSS	52 (81%)
HbS-β thalassemia	6 (9%)
HbSC	3 (5%)
Others	3 (5%)
# of blood transfusion in the preceding two years	2 (0–8)
# of vaso-occlusive episodes in the preceding two years	1 (0–2)
History of	
Acute thoracic syndrome	9 (14%)
Osteomyelitis	5 (8%)
Cholelithiasis or cholecystectomy	15 (23%)
Leg ulcers	0
Surgical splenectomy	7 (11%)
Auto-splenectomy	11 (17%)
TCD screening	35 (55%)
Normal	26 (74% *)
Conditional	7 (20% *)
Abnormal	2 (6% *)
Hydroxyurea therapy	45 (70%)
Haematopoietic stem cell transplantation	4 (6%)

Continuous variables reported median with interquartile ranges (IQR, 25th–75th percentile). Categorical variables reported as count (% of total). * % calculated of total TCD results analysed. #: numbers.

**Table 2 jcm-12-00007-t002:** Echocardiographic values of the cohort (*n* = 64).

Indexed LV mass (g) ^1^	44.87 ± 11.05
LV ejection fraction (%)	64 ± 4
LV diameter (mm)	42 ± 9
LV GLS (%)	23.2 ± 3.1
Indexed LA volume (mL/m^2^) ^1^	30.55 ± 8.84
Mitral E/e′ ratio	7.27 ± 1.32

Continuous variables are presented as mean ± SD. LV left ventricle, LA left atrial, GLS global longitudinal strain ^1^ See Methods section for LA volume and LV mass evaluation. Note: LA dilation was defined by LA volume/BSA^1.48^ derived Z scores [26]. LV hypertrophy was defined using a partition value of 45 g/(m^2.16+0.09^) [21,22]. LV dilation was defined by LV diameter mm/BSA derived Z scores [24].

**Table 3 jcm-12-00007-t003:** Clinical and laboratory data of study groups based on the presence of LAD.

	Left Atrial Dilatation (LAD)		
	Present (*n* = 41–65%)	Absent (*n* = 22–35%)	*p*
Gender (M%–F%)	M 59%–F 41%	M 55%–F 45%	0.76
Age	11 ± 7	9 ± 6	0.25
Hb (g/dL)	9.21 ± 0.83	9.80 ± 0.82	0.009
Absolute Reticulocyte count (10^4^/uL)	33.8 ± 15.29	25.8 ± 12.73	0.04
Creatinine (mg/dL)	0.37 ± 0.14	0.39 ± 0.12	0.511
Total bilirubin (mg/dL)	1.61 (1.25–2.52)	1.59 (0.84–2.52)	0.313
Unconjugated bilirubin (mg/dL)	1.39 (0.99–2.28)	0.92 (0.57–1.81)	0.58
ALT/SGPT (U/L)	33 ± 25	20 ± 15	0.009
AST/SGOT (U/L)	52 ± 25	42 ± 16	0.096
LDH (U/L)	700 ± 302	610 ± 324	0.274
GGT (U/L)	21 (17–32)	18 (13–23)	0.095
CRP (mg/dL)	1.07 (0.44–3.32)	0.93 (0.27–2.70)	0.459
HbS %	62.6 ± 14.02	56.0 ± 12	0.070
HbF %	11.3 ±7.1	13.4 ± 9.1	0.328
Nt-proBNP (pg/mL)	62.2 (42.9–116.0)	45.3 (22.3–79.1)	0.131
Ferritin ng/mL	463 (119–954)	302 (101–815)	0.555
N° of blood transfusions *	3 (1–9)	1 (0–6)	0.090
N° of pain crisis *	1 (0–2)	1 (0–2)	0.180

Continuous variables are presented as mean ± SD or as medians with interquartile ranges (IQR, 25th–75th percentile), as appropriate according to the distribution. Categorical variables are described using counts and percentages. * The number of blood transfusions and the number of pain crisis episodes reported were considered over the past two years.

**Table 4 jcm-12-00007-t004:** Clinical and laboratory data of study groups based on the presence of LVH.

	Left Ventricular Hypertrophy (LVH)		
	Present (*n* = 29–45%)	Absent (*n* = 35–55%)	*p*
Gender (M%–F%)	M 62%–F 38%	M 54%–F 46%	0.53
Age	11 ± 7	9 ± 6	0.27
Hb (g/dL)	9.11 ± 0.85	9.69 ± 0.80	0.006
Absolute Reticulocyte count (10^4^/uL)	35.61 ± 16.82	26.62 ± 12.10	0.031
Creatinine (mg/dL)	0.36 ± 0.09	0.40 ± 0.15	0.192
Total bilirubin (mg/dL)	2.09 ± 1.49	2.24 ± 1.86	0.733
Unconjugated bilirubin (mg/dL)	1.46 (1.01–2.66)	1.01 (0.55–1.56)	0.19
ALT/SGPT (U/L)	33 ± 22	25 ±21	0.073
AST/SGOT (U/L)	53 ± 24	45 ± 22	0.158
LDH (U/L)	721 ± 330	628 ± 287	0.233
GGT (U/L)	22 (18–29)	20 (14–28)	0.194
CRP (mg/dL)	0.94 (0.35–2.80)	1.13 (0.30–2.70)	0.609
HbS %	61.3 ± 14.9	59.4 ± 12.6	0.573
HbF %	10.5 ±7.48	13.2 ± 8.0	0.172
Nt-proBNP (pg/mL)	65.45 (39.35–131.83)	53.17 (36.22–86.68)	0.342
Ferritin (ng/mL)	463 (112–1655)	314 (110–610)	0.407
N° of blood transfusions *	5 (0–13)	1 (0–6)	0.096
N° of pain crisis *	1 (0–2)	1 (0–2)	0.880

Continuous variables are presented as mean ± SD or as medians with interquartile ranges (IQR, 25th–75th percentile), as appropriate according to the distribution. Categorical variables are described using counts and percentages. * The number of blood transfusions and the number of pain crisis episodes reported were considered over the past two years.

## Data Availability

The data presented in this study are available upon reasonable request from the corresponding author.

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
