# Peer review of "Echocardiographic Evaluation in Paediatric Sickle Cell Disease Patients: A Pilot Study"

_jcm, 2022, doi:10.3390/jcm12010007_

Round 1
Reviewer 1 Report
This is a single center cross sectional description of echocardiographic measurements in a cohort of children with sickle disease receiving care at an Italian center.
Some minor English is needed. Well written except for a few stray incorrect capitalizations, missing words, or misused words.
Additional details about the cohort and their treatment would be helpful.
How many are receiving chronic transfusion therapy for primary or secondary stroke prevention or severe complications of disease?
Are transcranial Doppler ultrasound screening performed in this group? Do any have signs of vasculopathy or increased velocities?
If none are receiving chronic monthly transfusions for stroke prevention or disease modification, the number of transfusions is quite high. Can a reason for the transfusions be provided?
For those taking hydroxyurea how long have they been taking it? What dosing approach is used at this hospital (i.e., low fixed, moderate fixed, escalated)? When is hydroxyurea typically started at this hospital? At birth? Only when complications occur?
The rate of surgical splenectomy in this group is high. Can the reasons for splenectomy be provided?
The prevalence of normal splenic function in this group is extraordinary. Can more detail be provided about normal? Is it possible that these children have reduced splenic function instead of normal splenic function?
The presentation of results needs some additional details. At one point they provide an IQR as a single numeral. The IQR is usually presented as the upper and lower quartile values. For the age, the median age is 10 years, so the IQR might be 4-12 or 6-14, etc. Please provide.
In the tables, please distinguish whether medians or means are used for values and please provide the standard deviation or the IQR so that there is some understanding of how spread out the values are.
In the figures, the values for absolute reticulocyte count has a misplaced comma, changing the values by an order of magnitude (i.e., they say 10,000-60,000 instead of 100,000 to 600,000).
In the figures, it would be helpful to display the median value in the box and whiskers plots.
In figure 1, it appears that the ALT is lower in the group with atrial dilatation, but I cannot tell if this is simply an illusion. Were the two groups flipped in this image?
In the tables, is PCR supposed to be CPR?
In table 2 and 3, are the number of blood transfusions and the number of vaso-occlusive episodes confined to a time period? In Table 1, it says the last two years, but no limitation is noted in Table 2 or 3.
Are lab values used a single previous value for each patient, or an average of values over the past two years? Were values during acute events excluded?
Reviewer 2 Report
The paper would be more effective if it were at least 50% shorter and if the introduction and discussion focused on the precise findings of the study.
The authors should give the size of the SCD pediatric population from which those having echocardiogram were selected. There should also be information regarding why the echocardiogram was done and whether it was during a hospitalization or when the patient was baseline.
The association of left ventricular and atrial sizes with lower hemoglobin and higher reticulocyte count is compatible with an association with degree of hemolysis. However, ALT and conjugated bilirubin are not good markers of hemolysis. Rather, LDH, AST and unconjugated bilirubin would be appropriate markers of hemolysis. The lack of consistent association with elevated LDH and AST is curious and could be commented on further.
